# Enhancing the Fertilizer Quality and Remediation Ability of Anaerobic Digestate via *Myrothecium verrucaria* Treatment

**Mingxin Yang [1],[†], Binbin Gong [2],[†], Jiayi Xu [1], Yonglin Sun [1], Pengjiao Tian [1],\* and Xiqing Wang [1],\***

[1]  College of Food Science Technology and Chemical Engineering, Hubei University of Arts and Science, Xiangyang 441053, China; ymx20030409@163.com (M.Y.); 18357913690@163.com (J.X.); sunyonglin2011@163.com (Y.S.)
[2]  College of Biological Sciences and Engineering, Xingtai University, Xingtai 054001, China; 201210644@xttc.edu.cn
\*  Correspondence: 13120018505@163.com (P.T.); xiqingwang91@163.com (X.W.)
†  These authors contributed equally to this work.

**Abstract:** Low fertilizer quality and remediation ability are considered the major factors hampering the land application of anaerobic digestate. Therefore, the role of *Myrothecium verrucaria* treatment in enhancing the fertilizer quality and remediation ability of digestate for land application was explored. Higher content of humic acid (7.5 g/L) with a higher degree of humification index and oxygen-containing functional groups was observed in the digestate receiving *Myrothecium verrucaria* treatment. Likewise, humic acid formed from *Myrothecium verrucaria* treatment had a higher capacity of heavy metal binding. Moreover, the viable and culturable count of *S. faecalis*, *S. typhi*, *C. perfringens*, *and E. coli* pathogens in the digestate decreased to approximately 12.50%, 41.70%, 18.87%, and 50.00% and 25.97%, 64.44%, 37.51%, and 75.27%, respectively, after treatment with *Myrothecium verrucaria*. This study provided a novel strategy to enhance the fertilizer quality, remediation ability, and biological safety of anaerobic digestate for land application.

**Keywords:** anaerobic digestate; humification; *Myrothecium verrucaria*; remediation; biological safety

## 1. Introduction

With the development of farming and the agricultural product processing industry on a large scale and the development of biogas as a clean energy source, biogas plants are developing towards a large scale and industrialization [1,2]. Biogas plants can produce clean energy while treating organic waste, and the treated digestate may also become organic fertilizer [3]. However, in practice, digestate is often not used directly in situ as expected, due to the low content of active compounds and high content of harmful substances (e.g., pathogens) [4]. Therefore, the disposal of digestate has become a major bottleneck to the industrialization of large-scale biogas plants.

Humification is a viable and promising strategy for resolving the issues associated with digestate disposal [5,6]. Humification is generally divided into the abiotic and biotic humification processes [6]. Different hypotheses, including the lignin–protein theory and sugar–amine condensation theory, have been used to explain humification [7,8]. Based on these theories, it was found that hydrous sodium manganese ore ($\delta$-MnO$_2$), which is widely present in soil and silt environments, can catalyze the formation of polyphenols, glucose, and glycine in a short time, with complex spectral characteristics similar to natural soil humic acids (HA), called humic acid-like substances [9]. Thus, the sugar–amine condensation theory and the polyphenol theory are linked to form a combined polyphenol–Maillard formation pathway, i.e., polysaccharides, polyphenols, and proteins can form humic acids through condensation reactions under the action of $\delta$-MnO$_2$, and it is speculated that this may be the most prevalent process of humic acid formation from organic matter in nature. Minerals from the environment, such as clays, Mn/Fe/Al/Si oxides, and $\delta$-MnO$_2$, function

as catalysts for abiotic humification processes, promoting the condensation of polyphenols and amino acids to generate HA [7,8]. Moreover, it was observed that the abiotic humification of organic materials (catechols and amino acids) could be accelerated by $\delta$-$MnO_2$, forming highly aromatic humic acids with strong heavy metal complexation and reduction capability [10]. Likewise, the structures of humic acids produced through abiotic humification processes comprise more carboxyl, hydroxyl, and quinone groups. During the biotic humidification process, key enzymes (such as laccase) released by fungi and actinomycetes oxidize organic monomers to quinone groups, which are then further oxidized and polymerized to synthesize HA [11]. Compared to the manganese dioxide-mediated abiotic humification process, the biotic humification process mediated by fungi plays the dual role of increasing humic acid content and reducing pathogenic bacteria. With its high laccase production, *Myrothecium verrucaria*, a significant fungus in the natural system, promotes the oxidative degradation of organic matter into humic acids. Likewise, it was revealed that *Myrothecium verrucaria* secretes a variety of active substances (e.g., quinolones) that inhibit the growth and reproduction of pathogenic bacteria. However, to date, the effect of *Myrothecium verrucaria* on digestate is still unclear, and further investigation is needed to enhance the active compounds and safety of application.

Therefore, to fill this information gap, the current study was undertaken to examine the ability of *Myrothecium verrucaria* to boost the humification of digestate. Moreover, excitation–emission matrix spectra (EEMs), Fourier transform infrared coupled with two-dimensional correlation spectroscopy (2D-COS FTIR) and electrochemical reduction, and oxidation methods were employed to characterize humic acids extracted from digestate. Additionally, the culturable and non-culturable count of pathogens before and after humification was determined through reverse transcription–quantitative PCR (RT-qPCR) to further assess the biological safety of digestate for land application.

## 2. Materials and Methods

### 2.1. Materials

Anaerobic digestate was collected from the biogas plant in the suburbs of Xiangyang, Hubei Province, China. This biogas plant uses a continuously stirred tank reactor operating under mesophilic conditions with a hydraulic retention time of 40 days. Chicken manure was used as the substrate for the reaction, feeding with 10% total solid content daily. The characteristics of anaerobic digestate are listed in Table 1. *Myrothecium verrucaria* used in this study was procured from Jilin Agricultural University, and was selected from the primary forest of Changbai Mountain and conserved in PDA broth culture and 25% glycerol stocks. Before the experiment, the strain was activated and selected on a solid PDA medium. Then, every single colony was peaked and inoculated into a PDA broth culture, which was then incubated overnight at 37 °C with shaking at 120 rpm. This process was repeated three times.

**Table 1.** Characteristics of the anaerobic digestate.

| Parameter | Symbol | Unit | Value |
| --- | --- | --- | --- |
| pH | / | / | $7.6 \pm 0.05$ |
| Total solids | TS | g/L | $4.3 \pm 0.5$ |
| Volatile solids | VS | g/L | $1.9 \pm 0.3$ |
| Soluble chemical oxygen demand | CODs | mg/L | $5010 \pm 352$ |
| Ammonical nitrogen | $NH_4^+$-N | mg/L | $4890 \pm 142$ |
| Free ammonia | $NH_3$-N | mg/L | $30 \pm 8$ |
| Orthophosphate | $PO_4^{-3}$-P | mg/L | $185 \pm 18$ |
| Lignin | / | g/L | $2.3 \pm 0.6$ |

### 2.2. Growth and Tolerance Study

The tolerance and growth characteristics of the *Myrothecium verrucaria* under varying concentrations of anaerobic digestate were carried out in 100 mL triangular flasks.

Briefly, the anaerobic digestate was first filtered to remove under-graded residues. The filtered digestate was diluted 0, 2, 4, 6, 8, and 10 times and sterilized in an oven (121 °C, 20 min). Different inoculum (1%, 2%, 3%) was added in different dilutions of digestates for inoculation and incubated in a shaking incubator (37 °C, 150 rpm). The experiment was terminated when there was no growth three consecutive times. Likewise, the biomass was measured at different times (0, 4, 8, 12, 24, 36, 48 h) by drying the microbial cells in an oven at 60 °C for 12 h and weighing. The highest tolerated concentration was obtained according to the tolerance and growth characteristics of the *Myrothecium verrucaria*. The biomass of *Myrothecium verrucaria* was measured following the method described previously [12]. Briefly, the fermentation broth derived from each time was first filtered, and then biomass was measured by drying the cells in a hot air oven at 60 °C overnight and weighing on an analytical balance. All experiments were performed in triplicate.

### 2.3. Batch Experiment Design

The batch experiment for the pretreatment of digestate was conducted in the highest tolerable concentration of microbial strain. Two experimental groups were set up in this study: (i) anaerobic digestate inoculated with microbial strain and (ii) without microbial strain. This experiment was conducted in an incubator at 37 °C for 16 days. Samples were taken after 0, 4, 8, 12, and 16 days to investigate the evolution of humic acid during the pretreatment process. A portion of the collected samples was used for humic acid extraction, structure determination, and safety and functional testing. The other portion was used to determine the evolution of the main components in the anaerobic digestate.

### 2.4. Extraction and Characterization of Humic Acid

Humic acid from the digestate samples was extracted and purified according to the standard method described in [13]. Briefly, 50 mL digestate was mixed with 0.1 M $Na_4P_2O_7$ and 0.1 M NaOH solution and shaken at 120 rpm for 12 h. Then, the mixture was centrifuged at 11,000 rpm for 20 min, followed by filtration through a 0.45 μm Millipore membrane. The supernatant was acidified to pH = 1 using 6 M HCl and left overnight. Subsequently, solid–liquid separation was performed via centrifugation at 5000 rpm for 10 min, and the supernatant was resuspended in a 100 mL mixture of NaOH and $Na_4P_2O_7$. The solution was centrifuged (5000 rpm, 10 min), and the supernatant was acidified to pH = 1 using 0.1 M HCl/0.3 M HF for 12 h. Finally, the mixture was centrifuged and percolated with distilled water to obtain the humic acid fraction.

### 2.5. Analysis Methods

The fluorescence characteristics of humic acid were determined through three-dimensional fluorescence spectrometry (Aqualog, HORIBA, Kyoto, Japan) following the method described in our already-published study [14]. Briefly, the excitation–emission matrix (EEM) spectra of collected digestate samples were obtained by employing three-dimensional fluorescence spectrometry with a wavelength range of 200–600 nm and increments of 2 nm and 5 nm. Then, the evolution of organic compounds in samples was quantified through the parallel factor (PARAFAC) analysis of EEMs using MATLAB R2019a (Mathworks, Natick, MA, USA) with the DOMFluor toolbox. Moreover, the fluorescence parameters, including the humification and biological indexes, were calculated using the PARAFAC analysis of EEMs. Likewise, the content of humic acid was determined according to the method described by [15]. The humic acid was characterized by employing Fourier transform infrared spectra (FTIR) coupled with two-dimensional correlation (2D-COS) spectra [15]. Briefly, humic acid derived from the aerobic digestate was first analyzed using the Nicolet IS10 FTIR spectrophotometer with a wavenumber range from 400 to 4000 cm$^{-1}$. Then, the 2D-COS analysis was performed on the data matrix obtained through FTIR using 2D-shige software [16].

### 2.6. Pathogen Analysis

According to our published study, *Escherichia coli*, *Clostridium perfringens*, *Streptococcus faecalis*, and *Salmonella typhimurium* were the main pathogenic species in the anaerobic digestates. Thus, the different states (culturable and non-cultivable (VBNC) of these pathogens in the anaerobic digestates before and after treatment were quantified following the method described in [4]. Briefly, reverse transcription–quantitative PCR was used to determine the total number of viable pathogens, while the culturable state was counted using a selective agar medium. The count of pathogens in the viable but non-culturable state (VBNC) was obtained by subtracting the number of culturable pathogen counts from the total number of pathogens in the viable state.

### 2.7. Heavy Metal Binding Analysis

The common heavy metal species copper (Cu) and chromium (Cr) were selected as representative metal ions to explore the heavy metal binding ability of HA according to the method described in [14]. Briefly, the HA fraction was diluted to a concentration of approximately 10 mg $L^{-1}$, and 0.1 M metal solution was added to a series of vials that contained 50 mL of diluted water. A series of experiments was performed with initial heavy metal concentrations that ranged from 5 to 100 μM. Then, the titrated solution was thoroughly mixed and left for 24 h at room temperature to ensure complexation equilibrium. All experiments were performed in triplicate to obtain statistically reliable results. Afterward, the titrated solutions were analyzed using spectral scanning to determine the fluorescence intensity. The results were used to simulate the Stern–Volmer model.

## 3. Results and Discussion

### 3.1. Growth and Tolerance of Myrothecium verrucaria under Varying Anaerobic Digestate Concentrations

The growth of *Myrothecium verrucaria* in different concentrations of anaerobic digestates was explored by determining its biomass yield (Figure 1a–c). The results showed that a higher degree of inhibition was observed in 0-fold and 2-fold dilutions of anaerobic digestates regardless of the inoculum, while the highest growth of *Myrothecium verrucaria* was observed in a 4-fold dilution of anaerobic digestates, where the biomass yield was 1.28 mg, 1.50 mg, and 1.51 mg in inoculum of 1%, 2%, and 3%, respectively. Moreover, it was perceived that no significant improvement in growth was observed in 6-fold, 8-fold, and 10-fold dilutions of the anaerobic digestate. Therefore, to treat digestates using *Myrothecium verrucaria*, a 4-fold dilution of the digestate and 2% inoculum are sufficient.

The tolerance of *Myrothecium verrucaria* to different concentrations of anaerobic digestates was assessed to understand its application potential. The results showed that *Myrothecium verrucaria* had a low tolerance to high concentrations of anaerobic digestates (0- and 2-times dilution). At the same time, no significant differences in tolerance were observed for anaerobic digestates diluted 4, 6, 8, and 10 times (Figure 1d–f). The maximum tolerance of *Myrothecium verrucaria* was found in the 4-fold dilution of the anaerobic digestate. Previous studies reported that the anaerobic digestates are not harmless because they contain heavy metals, free ammonia, pesticides, and pathogenic bacteria, which can significantly inhibit microbial or other organism's growth [17,18]. For instance, due to its small molecular size, ammonia can easily diffuse through cell membranes and accumulate in the cytoplasm, which may adversely affect cell physiology and metabolism [17]. In this study, the original content of free ammonia and ammonical nitrogen in the digestate was 4890 ± 142 and 30 ± 8 mg/L, respectively, which may inhibit the growth of *Myrothecium verrucaria*.

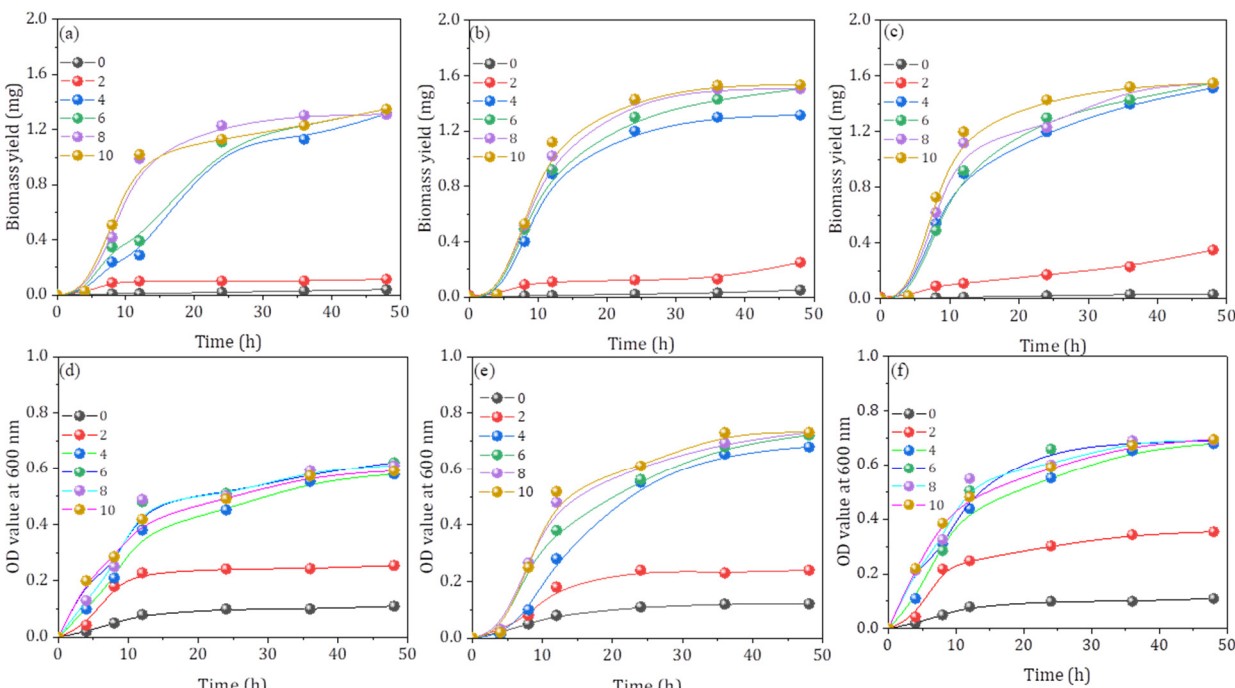

**Figure 1.** Tolerance and growth profile of *Myrothecium verrucaria* in the different inoculum conditions and different concentrations of anaerobic digestates. ((**a**,**d**): 1% inoculum; (**b**,**e**): 2% inoculum; (**c**,**f**): 3% inoculum).

### 3.2. Effect of Digestate Treatment on the Composition of Dissolved Organic Matter

Fluorescence EEM spectroscopy is often used to monitor the composition and evolution of dissolved organic matter (DOM) during the humification process [19,20]. This study observed a significant difference in the EEM spectra of digestates from *Myrothecium verrucaria* treatment (Figure 2a,b). Likewise, five fluorescent components (C1–C5) were characterized using PARAFAC analysis based on all EEM spectral data from anaerobic digestates. According to previous studies, components C1 and C3 (Ex/Em = 317 nm/405 nm and Ex/Em = 370 nm/415 nm, respectively) were ascribed to humic-like compounds [14,21], component C2 (Ex/Em = 407 nm/480 nm) was attributed to fulvic-like compounds [20], and components C4 (Ex/Em = 274 nm/305 nm) and C5 (Ex/Em = 280 nm/295 nm) were ascribed to tyrosine-like and tryptophan-like substances, respectively [14,21]. The contents of these five components in anaerobic digestates during the treatment process were monitored and labelled as $F_{max}$. As shown in Figure 2c, the $F_{max}$ values of components 1 and 3 increased steadily with the treatment time, whereas the $F_{max}$ of components 4 and 5 decreased during the treatment process. These results suggested that all treatment processes increased humic-like substances but decreased protein-like components. The increase in humic-like substances might be attributed to protein-like substances, as the precursor substances are quickly degraded and transformed into humic-like substances during the humification process [22]. Moreover, it was perceived that the highest contents of humic-like substances (63%) were observed in anaerobic digestate receiving *Myrothecium verrucaria* treatment. In contrast, lower contents of humic-like substances were found in the digestate without treatment (38%). This means that the use of *Myrothecium verrucaria* enhanced the degree of the humification process. Along with the transformation of DOM, the content of HA increased from 4.9 g/L to 7.5 g/L in the *Myrothecium verrucaria* treatment group.

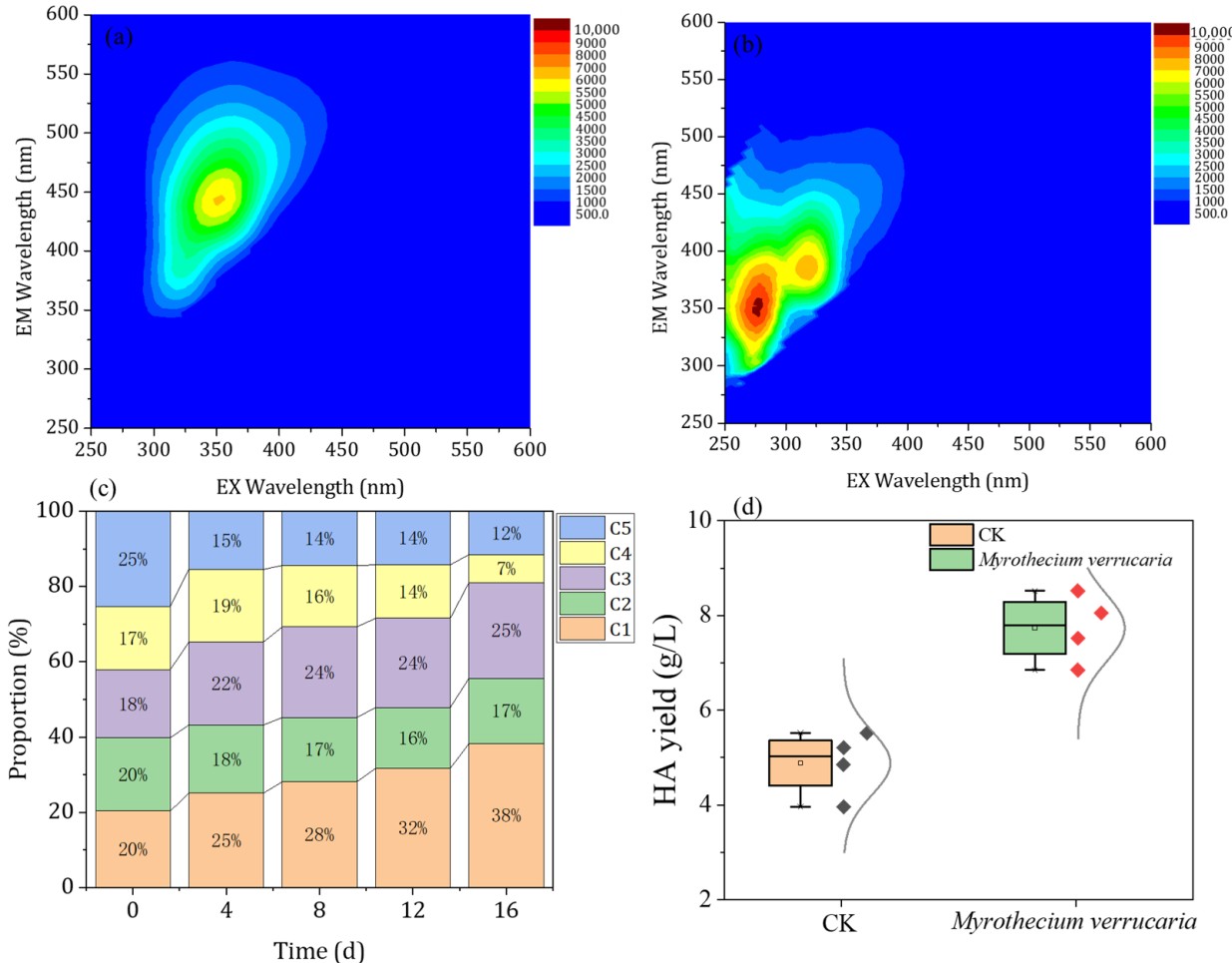

**Figure 2.** The excitation–emission matrix (EEM) spectra of digestate before (**a**) and after treatment (**b**). The relative proportion of excitation–emission matrix fluorescence components in the *Myrothecium verrucaria* treatment group (**c**). The humic acid content in control groups and the *Myrothecium verrucaria* treatment group (**d**). The data points in the graph are for parallel measurements).

Based on the EEM data, some fluorescence indices such as the fluorescence index (FI), humification index (HIX), and biological index (BIX) were calculated to reveal HA's sources and structure information. FI is an indicator of the HA source with higher (>1.4) and lower values (<1.4) indicating microbial and terrestrial sources, respectively [21]. In the *Myrothecium verrucaria* treatment groups, the value of FI firstly increased and then decreased during the 16 days of treatment (Figure 3a). This finding indicated that the humic acid formation process in the *Myrothecium verrucaria* treatment groups could be divided into two phases, with the first being mainly a microbial-dominated humification process, while the second phase was a joint microbial and redox-dominated humification process. BIX, as an important parameter, was correlated with the autochthonous biological activity of HA. Generally, a high BIX (>1.0) value corresponds to a predominantly autochthonous origin of HA and the presence of freshly produced HA [19,23]. The BIX values of the HA from the treatment groups decreased consistently, indicating that the generated HA showed an ageing trend with increasing treatment times. Moreover, the HIX indicates the degree of humification, which is positively related to the complexity of the HA structure [15]. The HIX values of HA in the treatment groups showed an increasing trend, indicating that the maturity or complexity of HA gradually increased with the treatment time [15,23]. Therefore, it is concluded that *Myrothecium verrucaria* contributed to the humification process of anaerobic digestate, thus improving its quality as fertilizer.

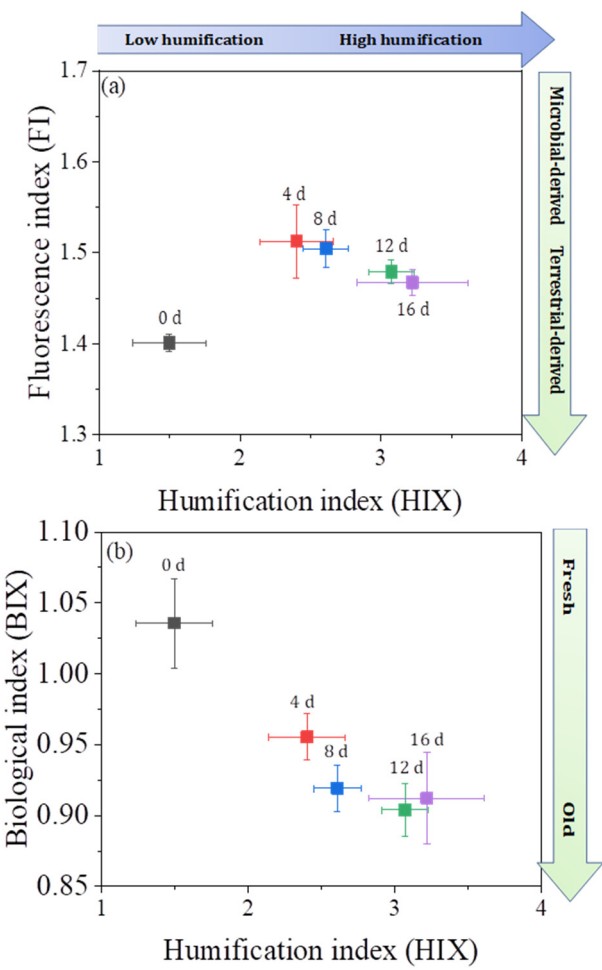

**Figure 3.** Correlations of humification index (HIX) with fluorescence index (FI) (**a**) and biological index (BIX) (**b**) of humic acid formed from the *Myrothecium verrucaria* treatment group.

### 3.3. Effect of Digestate Treatment on Structural Characteristics of Humic Acid

The potential of humic acids as fertilizer is significantly correlated to their structural characteristics; therefore, the variations in structural characteristics of humic acids under varying treatment conditions were determined using FTIR coupled with the 2D-COS analysis method. In the synchronous map spectra, significant differences between *Myrothecium verrucaria* treatment groups and the control group were observed (Figure 4a–c). In the *Myrothecium verrucaria* treatment group, six auto-peaks at 3400, 2900, 1805, 1650, 1230, and 1080 cm$^{-1}$ and eleven cross-peaks at (2900, 3400), (1080, 3400), (1805, 3400), (1650, 3400), (1080, 1805), (1650, 1805), (1230, 3400), (1230, 1805), (1230, 1650), (1080, 1650), and (1080, 1230) were observed. The peaks at 3400, 2900, 1805, 1650, 1230, and 1080 cm$^{-1}$ corresponded to the N-H (amides), C-H (aliphatic substances), C=C (ketones), C=O-OH (carboxylic acid, quinone), C-OH (phenol), and C=O (polysaccharides) functional groups, respectively. Furthermore, *Myrothecium verrucaria* for digestate treatment boosted the contents of oxygen-containing functional groups in the derived HA. The leading functional group responsible for its use as a remediation agent for contaminated soils was the existence of oxygen-containing functional groups (e.g., C=O-OH and C-OH) in the humic acid. Yang and Hodson (2019) confirmed that the oxygen-containing functional groups (e.g., carboxyl and hydroxyl) contributed to the adsorption of various heavy metals. Likewise, Xu et al. (2020) demonstrated that the quinone groups in the HA served as an electron-transferring shuttle, promoting various biological and biochemical processes for removing pollutants [24]. Thus, humic acids formed from the *Myrothecium verrucaria* treatment groups

may possess a more remarkable ability for the remediation of contaminated soils, which, in turn, enhances the usability of anaerobic digestate as fertilizer.

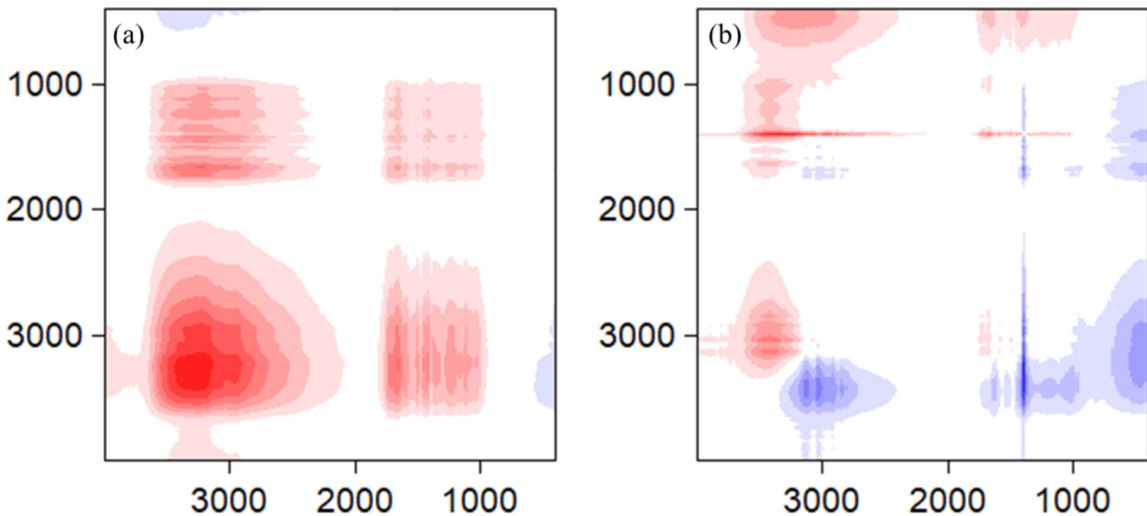

**Figure 4.** Synchronous (**a**) and asynchronous (**b**) 2D-COS FTIR spectra of humic acid formed from the *Myrothecium verrucaria* treatment group.

When compared with the synchronous spectrum, seven negative cross-peaks at (2900, 3400), (1080, 3400), (1650, 3400), (1230, 3400), (1080, 1805), (1230, 1805), and (1650, 1805) and three positive cross-peaks at (1080, 1650), (1805, 2900), and (1230, 1650) were observed in the asynchronous correlation spectrum of HA formed from the *Myrothecium verrucaria* treatment groups (Figure 4b),. These cross-peaks in the asynchronous map indicated that the surface functional groups of HA changed at different rates during the treatment process [15,25]. According to Noda's rules, the signs of the cross-peaks ($\Phi$ (V1, V2) > 0) and V1 > V2 suggest that the peaks in the V2 spectra appeared after the other peaks in the V1 spectra [16]. Conversely, when the signs of the cross-peaks were negative, a change in the peak in the V2 spectra appeared before the other peaks in the V1 spectra. Thus, the sequence of functional group changes in the HA formed from *Myrothecium verrucaria* treatment groups was as follows: 3300 cm$^{-1}$ (N-H) or 2900 cm$^{-1}$ (C-H) > 1650 cm$^{-1}$ (C=O-OH) > 1230 cm$^{-1}$ (C-OH) > 1080 cm$^{-1}$ (C=O) > 1805 cm$^{-1}$ (C=C). These results demonstrated that the carboxylic acid, quinuous, and phenol groups in humic acid changed faster under the *Myrothecium verrucaria* treatment, thus contributing to the potential of humic acid as a soil remediation and fertilization agent.

### 3.4. Effect of Digestate Treatment on Its Biological Safety for Land Application

Pathogens in viable, culturable, and non-culturable states (VBNC) in the anaerobic digestate were quantified before and after the *Myrothecium verrucaria* treatment of digestate (Figure 5). The total viable count of *Streptococcus faecalis (S. faecalis)*, *Salmonella typhimurium (S. typhi)*, *Clostridium perfringens (C. perfringens)*, and *Escherichia coli (E. coli)* in the digestate before *Myrothecium verrucaria* pretreatment was approximately 800 ± 60, 480 ± 50, 530 ± 68, and 1006 ± 80 cfu/g HA, respectively. Likewise, the culturable count was approximately 90.56–96.25% of the total viable pathogens, while 3.75–9.44% of viable pathogens were found to be in the VBNC state. However, the count of pathogens in viable, culturable, and VBNC states in the anaerobic digestate after *Myrothecium verrucaria* treatment was significantly different from their count in the digestate before treatment (Figure 5b). The results showed that the count of viable *S. faecalis, S. typhi, C. perfringens, and E. coli* in the digestate decreased by approximately 12.5%, 41.7%, 18.9%, and 50.0%, respectively, after *Myrothecium verrucaria* treatment. Likewise, the culturable counts of pathogens decreased by approximately 26.0%, 64.4%, 37.5%, and 75.3%, respectively. However, the count of pathogens in the VBNC state increased from 3.75–9.44% to 28.25–54.01% after the same

treatment. These outcomes indicated that *Myrothecium verrucaria* treatment enhanced the biological safety of anaerobic digestate for land application. *Myrothecium verrucaria* has high activity of extracellular cuticle-degrading enzymes, chitinases, proteinases, and lipases [25–27]. These enzymes have been shown to have high pathogenic inhibitory effects. Meanwhile, several quinolones are metabolically produced by *Myrothecium verrucaria* with significant antibacterial effects [28]. Therefore, it is concluded that *Myrothecium verrucaria* treatment might improve the biological safety of anaerobic digestate for its application in soil as a fertilizer.

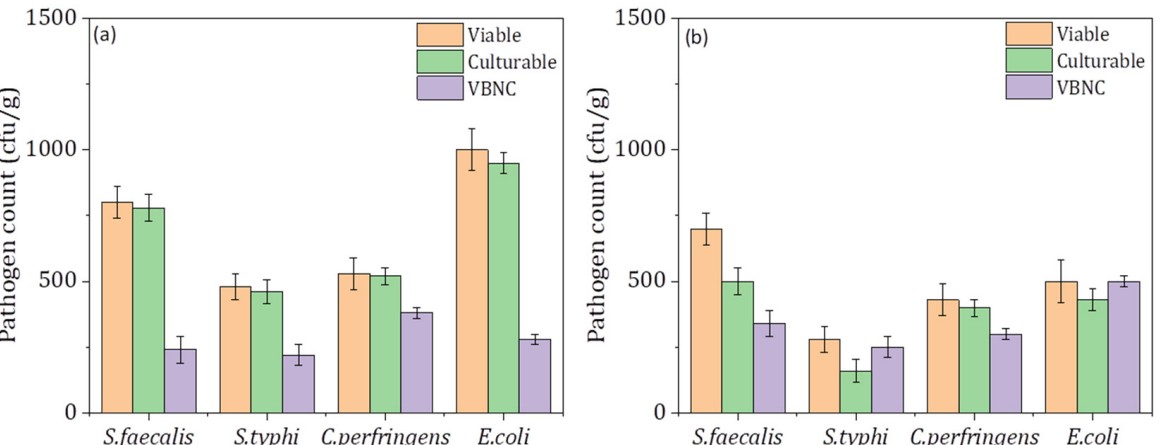

**Figure 5.** Quantification of *Streptococcus faecalis* (*S. faecalis*), *Salmonella typhimurium* (*S. typhi*), *Clostridium perfringens* (*C. perfringens*), and *Escherichia coli* (*E. coli*) into viable, culturable, and non-culturable (VBNC) states in humic acid before (**a**) and after (**b**) *Myrothecium verrucaria* pretreatment.

### 3.5. The Capacity of HA for Heavy Metal Remediation

The HAs collected from the control groups and *Myrothecium verrucaria* treatment groups were tested for their metal binding capacity towards target heavy metals (Cu and Cr). The fluorescence intensity dynamics that occur due to different heavy metal concentrates can be precisely described by the Stern–Volmer model (R 2 > 0.98, Figure 6). The results indicate that the differences in fluorescence intensities between the original HA (without heavy metal addition) and the samples after certain heavy metal addition can be used to represent the binding/adsorption capacity of HA with heavy metals [6,29]. Thus, in Figure 6, the higher decline in intensities suggests greater binding between HA and heavy metals. The results showed that the fluorescence intensity decreased by 37.2% and 41.6% when HA formed from the control group adsorbed Cu and Cr, respectively, while when HA derived from the *Myrothecium verrucaria* treatment groups adsorbed Cu and Cr, the fluorescence intensity decreased by 64.8% and 70.6%, respectively. These finding indicated that *Myrothecium verrucaria* treatment contributed to improving the remediation ability of heavy metals by HA.

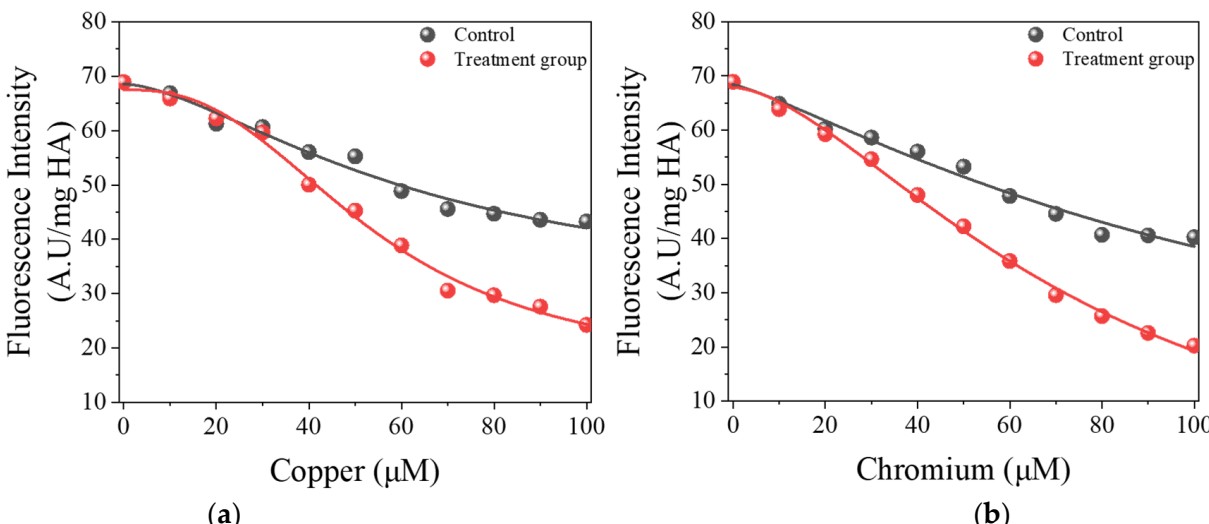

**Figure 6.** Variations in fluorescence intensities of humic acid derived from *Myrothecium verrucaria* treatment group with increasing initial concentrations of copper (**a**) and chromium (**b**).

## 4. Conclusions

This study explored the interactive influence of *Myrothecium verrucaria* treatment of digestate on the fertilizer quality and remediation ability of digestate for land application. The results showed that the *Myrothecium verrucaria* treatment process significantly increased the yield of HA derived from anaerobic digestate. Likewise, HA has a high content of oxygen-containing functional groups, resulting in the upgradation of digestate quality as fertilizer and remediation agent. Moreover, the qPCR results showed that *Myrothecium verrucaria* treatment significantly reduced the count of viable- and culturable-state pathogens, thus improving the biological safety of digestate for land application.

**Author Contributions:** Conceptualization: X.W. and M.Y.; Methodology, B.G. and J.X.; Software, M.Y. and Y.S.; Data curation, X.W. and P.T.; Writing-review and editing, X.W.; Funding acquisition, P.T. and X.W. All authors have read and agreed to the published version of the manuscript.

**Funding:** This work was supported by the Xiangyang Science and Technology Plan Project, Hubei, China (2021ABA003618) and the Foundation of Educational Commission of Hubei University of Arts and Science (QDF2021011 and QDF2022007).

**Institutional Review Board Statement:** Not applicable.

**Informed Consent Statement:** Not applicable.

**Data Availability Statement:** Data are available from the authors.

**Acknowledgments:** The authors gratefully acknowledge the support from the Xiangyang Science and Technology Plan Project, Hubei, China (2021ABA003618) and the Foundation of Education Commission of Hubei University of Arts and Science (QDF2021011 and QDF2022007).

**Conflicts of Interest:** The authors declare no conflict of interest.

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
