# Peer review of "Enhancing the Fertilizer Quality and Remediation Ability of Anaerobic Digestate via Myrothecium verrucaria Treatment"

_fermentation, doi:10.3390/fermentation9050484_

Round 1

Reviewer 1 Report

I would like to congratulate the authors for the quality and innovation of the research presented. I am certain that the results obtained are of great interest to the scientific community dedicated to studying the agronomic and environmental enhancement of biofertilizers.

Author Response

Response: Thanks for your comments. We are obliged to the reviewers of this manuscript, which contributed significantly to improve its quality.

Reviewer 2 Report

#1. Thanks to the author for well-scientifically executed data using modern precision equipment. I, however, raise the following concerns

(a). Chicken manure usually contains reasonably high humic acid-forming substances, so do you delineate this fact from your claimed influence by applied fungus?

(b).  Fertilizer quality goes beyond fortification of humic acid in the digestate, so what about the dynamics of other nutrients after pretreatment with the fungus i.e. P, N, and K? Would I be incorrect to think to thank some of these are lost (attenuated) during the complexation of metals with the humic substances enhanced and therefore non-bioavailable in soil for short rotation crops?

(c). Whats the economic projectile for the scaleup of these findings to benefit poultry farms that would sell the fortified digestate?

Author Response

(a). Chicken manure usually contains reasonably high humic acid-forming substances, so do you delineate this fact from your claimed influence by applied fungus?

Response: The original HA was not distinguished from the newly formed HA in this study. In the future, we will further analysis the different of structure and function between original HA and formed HA. Actually, this study explored the effect of Myrothecium verrucaria on the HA generation from anaerobic digestate. The chicken manure usually contains various precursors for humic acid-forming, such as aromatic compounds, amide compounds and aliphatic compounds. Generally, these complex organic compounds could first be decomposed into small-molecule organics and then transformed into recalcitrant macromolecular organic products through the repolymerization to form HA under relevant microorganism functions. Thus, the HA formed by Myrothecium verrucaria treatment is higher than the control group (7.5 g/L and 4.9 g/L, respectively).

(b). Fertilizer quality goes beyond fortification of humic acid in the digestate, so what about the dynamics of other nutrients after pretreatment with the fungus i.e. P, N, and K? Would I be incorrect to think to thank some of these are lost (attenuated) during the complexation of metals with the humic substances enhanced and therefore non-bioavailable in soil for short rotation crops?

Response: Thanks for your suggestion. The change of other nutrients (e.g., N, P and K) was not analysis in this study. This study only focuses on the effect of Myrothecium verrucaria on the himification of anaerobic digestion. In the future, the dynamic of other nutrients (e.g., N, P and K) during the humification process will be determined. Moreover, whether the utilization of other nutrients is affected during the complexation of metals with humic substances still needs to be further investigated.

(c). Whats the economic projectile for the scaleup of these findings to benefit poultry farms that would sell the fortified digestate?

Response: According to the report of Chinese National Development and Reform Commission, there are more than 120,000 large biogas projects in China, producing 25 billion m3 of biogas and about 30 million tons/year of digestate (Ministry of Agriculture, 2017). Agricultural use as fertilizer, soil improver, and growing medium is the straightforward and economically feasible digestate valorization route. Although the land application of digestate is a commonly used practice, the existence of contaminants (heavy metals) and pathogens necessitates further management of digestate. The current study demonstrated a proof-of-concept for anaerobic digestate treatment and resource recovery. According to the technical route of this experiment, more than 267000 tons/year of humic acid can be produced from the anaerobic digestate currently generated in China. The humic acid formed from the anaerobic digestate contains a higher content of oxygen-containing functional, contributing to its higher contaminant removal potential. Moreover, Myrothecium verrucaria-mediated biotic humification process can not only promote HA content but also reduce pathogenic counts in anaerobic digestate, thus making it a biologically safe fertilizer for land application. However, more studies are required at the pilot scale to test the feasibility and potential of the strategy for application at the industrial scale.

Reference:

Ministry of Agriculture, 2017. The People’s Republic of China. http://www.gov.cn/

xinwen/2017-02/10/content_5167076.htm.